# Progressive Weight Pruning of Deep Neural Networks using ADMM

## Abstract

Deep neural networks (DNNs), although achieving human-level performance in many domains, have very large model size that hinders their broader applications on edge computing devices. Extensive research work has been conducted on DNN model compression or pruning. However, most of the previous work has taken heuristic approaches. This work proposes a progressive weight pruning approach based on ADMM (Alternating Direction Method of Multipliers), a powerful technique to deal with non-convex optimization problems with potentially combinatorial constraints. Motivated by dynamic programming, the proposed method reaches extremely high pruning rate by using partial prunings with moderate pruning rates. Therefore, it resolves the accuracy degradation and long convergence time problems when pursuing extremely high pruning ratios. It achieves up to $34\times$ pruning rate for ImageNet data set and $167\times$ pruning rate for MNIST data set, significantly higher than those reached by existing work in the literature. Under the same number of epochs, the proposed method also achieves better convergence and higher compression rates. The codes and pruned DNN models are avilable in the link: `bit.ly/2zxdlss`.

## 1 Introduction

Deep neural networks (DNNs) have achieved human-level performance in many application domains such as image classification (Krizhevsky et al., 2012), object recognition (LeCun et al., 1998; He et al., 2016), natural language processing (Hinton et al., 2012; Dahl et al., 2012), etc. At the same time, the networks are growing deeper and bigger for higher classification/recognition performance (i.e., accuracy) (Simonyan & Zisserman, 2015). However, the very large DNN model size increases the computation time of the inference phase. To make matters worse, the large model size hinders DNN' deployments on edge computing, which provides the ubiquitous application scenarios of DNNs besides cloud computing applications.

As a result, extensive research efforts have been devoted to *DNN model compression*, in which DNN *weight pruning* is a representative technique. Han et al. (2015) is the first work to present the DNN weight pruning method, which prunes the weights with small magnitudes and retrains the network model, heuristically and iteratively. After that, more sophisticated heuristics have been proposed for DNN weight pruning, e.g., incorporating both weight pruning and growing (Guo et al., 2016), $L_1$ regularization (Wen et al., 2016), and genetic algorithms (Dai et al., 2017). Other improvement directions of weight pruning include trading-off between accuracy and compression rate, e.g., *energy-aware pruning* (Yang et al., 2017), incorporating regularity, e.g., *channel pruning* (He et al., 2017), and *structured sparsity learning* (Wen et al., 2016).

While the weight pruning technique explores the redundancy in the number of weights of a network model, there are other sources of redundancy in a DNN model. For example, the weight quantization (Leng et al., 2017; Park et al., 2017; Zhou et al., 2017; Lin et al., 2016; Wu et al., 2016; Rastegari et al., 2016; Hubara et al., 2016; Courbariaux et al., 2015) and clustering (Zhu et al., 2017; Han et al., 2016) techniques explore the redundancy in the number of bits for weight representation. The activation pruning technique (Jung et al., 2018; Sharify et al., 2018) leverages the redundancy in the intermediate results. While our work focuses on weight pruning as the major DNN model compression technique, it is orthogonal to the other model compression techniques and might be integrated under a single ADMM-based framework for achieving more compact network models.

The majority of prior work on DNN weight pruning take heuristic approaches to reduce the number of weights as much as possible, while preserving the expressive power of the DNN model. Then one may ask, how can we push for the utmost sparsity of the DNN model without hurting accuracy? and what is the maximum compression rate we can achieve by weight pruning? Towards this end, Zhang et al. (2018b) took a tentative step by proposing an optimization-based approach that leverages ADMM (Alternating Direction Method of Multipliers), a powerful technique to deal with non-convex optimization problems with potentially combinatorial constraints. This direct ADMM-based weight pruning technique can be perceived as a smart DNN regularization where the regularization target is dynamically changed in each ADMM iteration. As a result it achieves higher compression (pruning) rate than heuristic methods.

Inspired by Zhang et al. (2018b), in this paper we propose a progressive weight pruning approach that incorporates both an ADMM-based algorithm and masked retraining, and takes a progressive means targeting at extremely high compression (pruning) rates with negligible accuracy loss. The contributions of this work are summarized as follows:

- We make a key observation that when pursuing extremely high compression rates (say $150\times$ for LeNet-5 or $30\times$ for AlexNet), the direct ADMM-based weight pruning approach (Zhang et al., 2018b) cannot produce exactly sparse models upon convergence, in that many weights to be pruned are close to zero but not exactly zero. Certain accuracy degradation will result from this phenomenon if we simply set these weights to zero.

- We propose and implement the progressive weight pruning paradigm that reaches an extremely high compression rate through multiple partial prunings with progressive pruning rates. This progressive approach, motivated by dynamic programming, helps to mitigate the long convergence time by direct ADMM pruning.

- Extensive experiments are performed by comparing with many state-of-the-art weight pruning approaches and the highest compression rates in the literature are achieved by our progressive weight pruning framework, while the loss of accuracy is kept negligible. Our method achieves up to $34\times$ pruning rate for the ImageNet data set and $167\times$ pruning rate for the MNIST data set, with virtually no accuracy loss. Under the same number of epochs, the proposed method achieves notably better convergence and higher compression rates than prior iterative pruning and direct ADMM pruning methods.

We provide codes (both Caffe and TensorFlow versions) and pruned DNN models (both for the ImageNet and MNIST data sets) in the link: `bit.ly/2zxdlss`.

## 2 THE PROGRESSIVE WEIGHT PRUNING FRAMEWORK OF DNNs

This section introduces the proposed progressive weight pruning framework using ADMM. Section 2.1 describes the overall framework. Section 2.2 discusses the ADMM-based algorithm for DNN weight pruning (Zhang et al., 2018b), which we will improve and incorporate into the progressive weight pruning framework. Section 2.3 proposes a direct improvement of masked retraining to restore accuracy. Section 2.4 provides the motivations and details of the proposed progressive weight pruning framework.

### 2.1 THE OVERALL FRAMEWORK

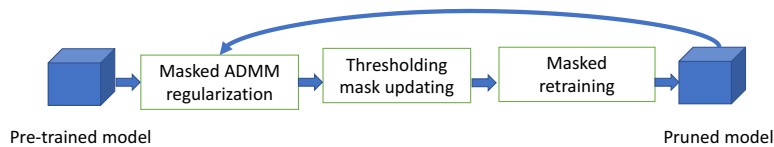

Figure 1: The overall progressive weight pruning framework including masked ADMM-based algorithm, thresholding mask updating, and masked retraining steps.

The overall framework of progressive weight pruning is shown in Figure 1. It applies the ADMM-based pruning algorithm on a pre-trained (uncompressed) network model. Then it defines thresholding masks, with which the weights smaller than thresholds are forced to be zero. To restore accuracy, the masked retraining step is applied, that only updates nonzero weights specified by the thresholding masks. The ADMM-based algorithm, thresholding mask updating, and masked retaining steps are performed for several rounds, and each round is considered as a partial pruning, progressively pushing for the utmost of the DNN model pruning. Note that in our progressive weight pruning framework, we change the ADMM-based algorithm into a "masked" version that reuses the partially pruned model by masking the gradients of the pruned weights, thereby preventing them from recovering to nonzero weights and thus accelerating convergence.

## 2.2 ADMM-based Pruning Algorithm

Our ADMM-based pruning algorithm takes a pre-trained network as the input and outputs a pruned network model satisfying some sparsity constraints. Consider an $N$-layer DNN, where the collection of weights in the $i$-th (convolutional or fully-connected) layer is denoted by $\mathbf{W}_i$ and the collection of biases in the $i$-th layer is denoted by $\mathbf{b}_i$. The loss function associated with the DNN is denoted by $f\big(\{\mathbf{W}_i\}_{i=1}^N, \{\mathbf{b}_i\}_{i=1}^N\big)$.

The DNN weight pruning problem can be formulated as:

$$\begin{aligned} \underset{\{\mathbf{W}_i\},\{\mathbf{b}_i\}}{\text{minimize}} \quad & f\big(\{\mathbf{W}_i\},\{\mathbf{b}_i\}\big), \\ \text{subject to} \quad & \mathbf{W}_i \in \mathbf{S}_i, \ i=1,\ldots,N, \end{aligned} \tag{1}$$

where $\mathbf{S}_i = \{\mathbf{W}_i \mid \text{card}(\mathbf{W}_i) \leq l_i\}, i=1,\ldots,N$ and $l_i$ is the desired number of weights in the $i$-th layer of the DNN. It is clear that $\mathbf{S}_1,\ldots,\mathbf{S}_N$ are nonconvex sets, and it is in general difficult to solve optimization problems with nonconvex constraints.

The problem can be equivalently rewritten in a format without constraints, namely

$$\underset{\{\mathbf{W}_i\},\{\mathbf{b}_i\}}{\text{minimize}} \quad f\big(\{\mathbf{W}_i\},\{\mathbf{b}_i\}\big) + \sum_{i=1}^N g_i(\mathbf{W}_i), \tag{2}$$

where $g_i(\cdot)$ is the indicator function of $\mathbf{S}_i$, i.e.,

$$g_i(\mathbf{W}_i) = \begin{cases} 0 & \text{if card}(\mathbf{W}_i) \leq l_i, \\ +\infty & \text{otherwise.} \end{cases} \tag{3}$$

The ADMM technique (Boyd et al., 2011) can be applied to solve the weight pruning problem by formulating it as:

$$\begin{aligned} \underset{\{\mathbf{W}_i\},\{\mathbf{b}_i\}}{\text{minimize}} \quad & f\big(\{\mathbf{W}_i\},\{\mathbf{b}_i\}\big) + \sum_{i=1}^N g_i(\mathbf{Z}_i), \\ \text{subject to} \quad & \mathbf{W}_i = \mathbf{Z}_i, \ i=1,\ldots,N. \end{aligned}$$

Through the augmented Lagrangian, the ADMM technique decomposes the weight pruning problem into two subproblems, and solving them iteratively until convergence. The first subproblem is:

$$\underset{\{\mathbf{W}_i\},\{\mathbf{b}_i\}}{\text{minimize}} \quad f\big(\{\mathbf{W}_i\},\{\mathbf{b}_i\}\big) + \sum_{i=1}^N \frac{\rho_i}{2}\|\mathbf{W}_i - \mathbf{Z}_i^k + \mathbf{U}_i^k\|_F^2. \tag{4}$$

This subproblem is equivalent to the original DNN training plus an $L_2$ regularization term, and can be effectively solved using stochastic gradient descent with the same complexity as the original DNN training. Note that we cannot prove global optimality of the solution to subproblem (4), just as we cannot prove optimality of the solution to the original DNN training problem.

On the other hand, the second subproblem is:

$$\underset{\{\mathbf{Z}_i\}}{\text{minimize}} \quad \sum_{i=1}^N g_i(\mathbf{Z}_i) + \sum_{i=1}^N \frac{\rho_i}{2}\|\mathbf{W}_i^{k+1} - \mathbf{Z}_i + \mathbf{U}_i^k\|_F^2.$$

Since $g_i(\cdot)$ is the indicator function of the set $\mathbf{S}_i$, the globally optimal solution to this subproblem can be explicitly derived as in Boyd et al. (2011):

$$\mathbf{Z}_i^{k+1} = \mathbf{\Pi}_{\mathbf{S}_i}(\mathbf{W}_i^{k+1} + \mathbf{U}_i^k), \qquad (5)$$

where $\mathbf{\Pi}_{\mathbf{S}_i}(\cdot)$ denotes the Euclidean projection onto the set $\mathbf{S}_i$.

Note that $\mathbf{S}_i$ is a nonconvex set, and computing the projection onto a nonconvex set is a difficult problem in general. However, the special structure of $\mathbf{S}_i = \{\mathbf{W}_i \mid \mathrm{card}(\mathbf{W}_i) \leq l_i\}$ allows us to express this Euclidean projection analytically. Namely, the optimal solution (5) is to keep the $l_i$ largest elements of $\mathbf{W}_i^{k+1} + \mathbf{U}_i^k$ and set the rest to zero (Boyd et al., 2011). Here we introduce set $\mathbf{P}_i$ for weights that were pruned to be zero. In every layer, $\mathbf{P}_i$ is a subset of $\mathbf{S}_i$. By introducing set $\mathbf{P}_i$, we introduce progressive pruning. We will show more detail in Algorithm 1.

Finally, we update the dual variable $\mathbf{U}_i$ as

$$\mathbf{U}_i^{k+1} = \mathbf{U}_i^k + \mathbf{W}_i^{k+1} - \mathbf{Z}_i^{k+1}. \qquad (6)$$

This concludes one iteration of the ADMM.

In the context of deep learning, the ADMM-based algorithm for DNN weight pruning can be understood as a smart DNN regularization technique (see Eqn. (4)), in which the regularization target (in the $L_2$ regularization term) is dynamically updated in each ADMM iteration. This is one reason that the ADMM-based algorithm for weight pruning achieves higher performance than heuristic methods and other regularization techniques (Wen et al., 2016), and the Projected Gradient Descent technique (Zhang et al., 2018a).

### 2.3 MASKED RETRAINING STEP

Applying the ADMM-based pruning algorithm alone has limitations for high compression rates. At convergence, the pruned DNN model will not be exactly sparse, in that many weights to be pruned will be close to zero instead of being exactly equal to zero. This is due to the non-convexity of Subproblem 1 in the ADMM-based algorithm. Certain accuracy degradation will result from this phenomenon if we simply set those weights to zero. This accuracy degradation will be non-negligible for high compression rates.

Instead of waiting for the full convergence of the ADMM-based algorithm, a masked retraining step is proposed, that (i) terminates the ADMM iterations early, (ii) keeps the $l_i$ largest weights (in terms of magnitude) and sets the other weights to zero, and (iii) performs retraining on the nonzero weights (with zero weights masked) using the training data set. More specifically, masks are applied to gradients of zero weights, preventing them from updating. Essentially, the ADMM-based algorithm sets a good starting point, and then the masked retraining step encourages the remaining nonzero weights to learn to recover classification accuracies.

Integrating masked retraining after the ADMM-based algorithm, a good compression rate can be achieved with reasonable training time. For example, we can achieve $21\times$ model pruning rate without accuracy loss for AlexNet using a total of 417 epochs, much faster than the iterative weight pruning method of Han et al. (2016), which achieves $9\times$ pruning rate in a total of 960 epochs. When translating into training time, our time of training is 72 hours using single NVIDIA 1080Ti GPU, whereas the reported training time in Han et al. (2016) is 173 hours.

### 2.4 PROGRESSIVE WEIGHT PRUNING

The algorithm for producing an intermediate model is discussed in Algorithm 1. Although the ADMM-based pruning algorithm in Section 2.2 and the masked retraining step in Section 2.3 together can achieve the state-of-the-art model compression (pruning) rates for many network models, we find limitations to this approach at extremely high pruning rates, for example at $150\times$ pruning rate for LeNet-5 or $30\times$ pruning rate for AlexNet.

Specifically, with a very high weight pruning rate, it takes a relatively long time for the ADMM-based algorithm to choose which weights to prune. For example, it is difficult for the ADMM-based algorithm to converge for $30\times$ pruning rate on AlexNet but easy for $21\times$ pruning rate.

To overcome this difficulty, we propose the progressive weight pruning method. This technique is motivated by dynamic programming, achieving a high weight pruning rate by using partial pruning models with moderate pruning rates. We use Figure 2 as an example to show the process used to achieve $30\times$ weight pruning rate in AlexNet without accuracy loss. In Figure 2 (a), we start from three partial pruning models, with $15\times$, $18\times$, and $21\times$ pruning rates, which can be directly derived from the uncompressed DNN model via the ADMM-based algorithm with masked retraining. To achieve $24\times$ weight pruning rate, we start from these three models and check which gives the highest accuracy (suppose it is the $15\times$ one). Because we start from partial pruning models, the convergence rate is fast. We then replace $15\times$ partial pruning model by $24\times$ model to derive the $27\times$ model, see Figure 2 (b). In this way we always maintain three partial results and limit the total searching time. Suppose this time the $18\times$ pruning model results in the highest accuracy and then we replace it with the $27\times$ one. Finally, in Figure 2 (c), we find $24\times$ model gives highest accuracy to reach $30\times$ pruning rate.

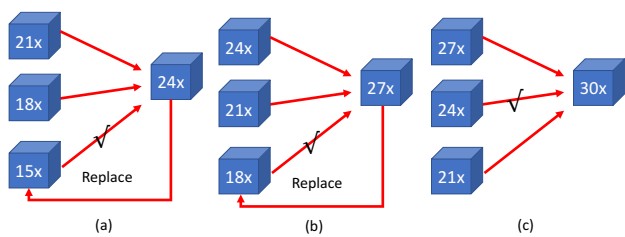

Figure 2: Illustration of the progressive weight pruning algorithm.

Note that during progressive weight pruning, to leverage the partial pruning models, we use "masked" training when we reuse the partial pruning models in the ADMM-based algorithm. Specifically, it masks the gradients of the already pruned weights to prevent them from recovering to nonzero values. In this way, the algorithm is encouraged to focus on pruning nonzero weights.

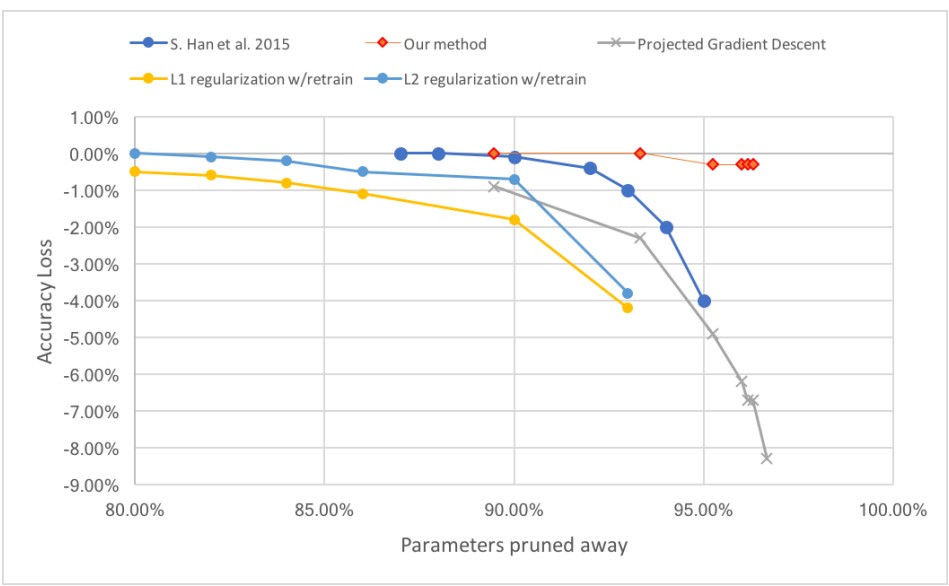

Figure 3: Results on AlexNet for ImageNet dataset. When pursuing high pruning rates, multiple pruning methods suffer severely from the degradation in Top-5 accuracy. The proposed method mitigates the issue even in extremely high pruning rates.

Figure 3 demonstrates that pruning gets harder when pursuing high pruning rates. Methods such as Projected Gradient descents and Iterative Pruning (Han et al., 2015) occurs large accuracy loss when pruning rates are high. However, the proposed method mitigates the performance degradation.

**Algorithm 1** Progressive weight pruning using ADMM
_______________________________________________________________
    Define constant Max-Iterations
    Define constant ADMM-Iterations, empirically 1/10 of Max-Iterations
    **for** $i$ in number of layers  **do**
        Apply mask on gradients for weights in $\mathbf{P}_i$
    **end for**
    **for** each $k$ in Max-Iterations **do**
        **for** $i$ in the number of layers  **do**
            Solve (Eqn.4) and update $\mathbf{W}_i$'s and $\mathbf{b}_i$'s;
            **if** k is dividable by ADMM-Iterations **then**:
                Update $\mathbf{Z}_i$'s by performing Euclid mapping (Eqn.5);
                Update $\mathbf{U}_i$'s according to dual update (Eqn.6).
            **end if**
        **end for**
    **end for**
    **for** $i$ in number of layers  **do**
        Apply masks on elements of $\mathbf{W}_i$;
        Apply masks on gradients of $\mathbf{W}_i$;
    **end for**
    Apply masked retrain
_______________________________________________________________

## 3  EXPERIMENTAL RESULTS AND DISCUSSIONS

Table 1: Comparisons of weight pruning results on AlexNet for ImageNet data set.

| Method | Top-5 Acc. | No. Para. | Rate |
|---|---|---|---|
| Uncompressed | 80.27% | 61.0M | 1× |
| Network Pruning (Han et al., 2015) | 80.3% | 6.7M | 9× |
| Optimal Brain Surgeon (Dong et al., 2017) | 80.0% | 6.7M | 9.1× |
| Low Rank and Sparse Decomposition (Yu et al., 2017) | 80.3% | 6.1M | 10× |
| Fine-Grained Pruning (Mao et al., 2017) | 80.4% | 5.1M | 11.9× |
| NeST (Dai et al., 2017) | 80.2% | 3.9M | 15.7× |
| Dynamic Surgery (Guo et al., 2016) | 80.0% | 3.4M | 17.7× |
| ADMM Pruning (Zhang et al., 2018b) | 80.2% | 2.9M | 21× |
| **Progressive Weight Pruning** (BVLC Model) | 80.2% | 2.02M | 30× |
| **Progressive Weight Pruning** (BVLC Model) | 80.0% | 1.97M | 31× |
| **Progressive Weight Pruning** (CaffeNet Model) | 80.2% | 2.02M | 30× |
| **Progressive Weight Pruning** (CaffeNet Model) | 80.0% | 1.97M | 31× |

Table 2: Top-5 accuracy of direct ADMM pruning (Zhang et al., 2018b) and progressive pruning at different pruning rates on AlexNet for ImageNet data set.

| Pruning Rate | Direct ADMM Pruning | Progressive Weight Pruning |
|---|---|---|
| 18× | 80.3% | 80.9% |
| 21× | 80.2% | 80.8% |
| 30× | 76.7% | 80.2% |

### 3.1  EXPERIMENTAL SETUPS

We evaluate the proposed ADMM-based progressive weight pruning framework on the ImageNet ILSVRC-2012 data set (Deng et al., 2009) and MNIST data set (LeCun et al., 1998). We also use DNN weight pruning results from many previous works for comparison. For ImageNet data set, we test on a variety of DNN models including AlexNet (both BAIR/BVLC model and CaffeNet model), VGG-16, and ResNet-50 models. We test on LeNet-5 model for MNIST data set. The accuracies of the uncompressed DNN models are reported in the tables for reference.

Table 3: Comparisons of weight pruning results on VGG-16 for ImageNet data set.

| Method | Top-5 Acc. | No. Para. | Rate |
|---|---|---|---|
| Uncompressed | 88.7% | 138M | 1× |
| Network Pruning (Han et al., 2015) | 89.1% | 10.6M | 13× |
| Optimal Brain Surgeon (Dong et al., 2017) | 89.0% | 10.3M | 13.3× |
| Low Rank and Sparse Decomposition (Yu et al., 2017) | 89.1% | 9.2M | 15× |
| ADMM Pruning (Zhang et al., 2018b) | 88.7% | 7.26M | 19.5× |
| **Progressive Weight Pruning** | 88.7% | 4.6M | 30× |
| **Progressive Weight Pruning** | 88.2% | 4.1M | 34× |

Table 4: Comparisons of weight pruning results on ResNet-50 for ImageNet data set.

| Method | Top-5 Acc. | No. Para. | Rate |
|---|---|---|---|
| Uncompressed | 92.40% | 25.6M | 1× |
| Fine-grained Pruning (Mao et al., 2017) | 92.3% | 9.8M | 2.6× |
| **Progressive Weight Pruning** | 92.3% | 4.3M | 6× |
| **Progressive Weight Pruning** | 92.1% | 2.8M | 9.16× |
| **Progressive Weight Pruning** | 91.5% | 1.47M | 17.43× |

We implement our codes in Caffe (Jia et al., 2014). Experiments are tested on 12 Nvidia GTX 1080Ti GPUs and 12 Tesla P100 GPUs. As the key parameters in ADMM-based weight pruning, we set the ADMM penalty parameter $\rho$ to $1.5 \times 10^{-3}$ for the masked ADMM-based algorithm. When targeting at a high weight pruning rate, we change it to $3.0 \times 10^{-3}$ for higher performance. To eliminate the already pruned weights in partial pruning results from the masked ADMM-based algorithm, $\rho_i$ is forced to be zero if no more pruning is performed for a specific layer $i$. We use an initial learning rate of $1.0 \times 10^{-3}$ for the masked ADMM-based algorithm and an initial learning rate of $1.0 \times 10^{-2}$ for masked retraining.

We provide the codes (both Caffe and TensorFlow versions) and all pruned DNN models (both for ImageNet and MNIST data sets) in the link: `bit.ly/2zxdlss`.

## 3.2 Comparison Results and Discussions

Table 1 presents the weight pruning comparison results on the AlexNet model between our proposed method and prior works. Our weight pruning results clearly outperform the prior work, in that we can achieve 31× weight reduction rate without loss of accuracy. Our progressive weight pruning also outperforms the direct ADMM weight pruning in Zhang et al. (2018b) that achieves 21× compression rate. Also the CaffeNet model results in slightly higher accuracy compared with the BVLC AlexNet model. Table 2 presents more comparison results with the direct ADMM pruning. It can be observed that (i) with the same compression rate, our progressive weight pruning outperforms the direct pruning in accuracy; (ii) the direct ADMM weight pruning suffers from significant accuracy drop with high compression rate (say 30× for AlexNet); and (iii) for a good compression rate (18× and 21×), our progressive weight pruning technique can even achieve higher accuracy compared with the original, uncompressed DNN model.

Table 3, Table 4, and Table 5 present the comparison results on the VGG-16, ResNet-50, and LeNet-5 (for MNIST) models, respectively. These weight pruning results we achieved clearly outperform the prior work, consistently achieving the highest sparsities in the benchmark DNN models. On the VGG-16 model, we achieve 30× weight pruning with comparable accuracy with prior works, while the highest pruning rate in prior work is 19.5×. We also achieve 34× weight pruning with minor accuracy loss. For ResNet-50 model, we have tested 17.43× weight pruning rate and confirmed minor accuracy loss. In fact, there is limited prior work on ResNet weight pruning for ImageNet data set, due to (i) the difficulty in weight pruning since ResNet mainly consists of convolutional layers, and (ii) the slow training speed of ResNet. Our method, on the other hand, achieves a relatively high training speed, thereby allowing for the weight pruning testing on different large-scale DNN models.

Table 5: Comparisons of weight pruning results on LeNet-5 for MNIST data set.

| Method | Accuracy | No. Para. | Rate |
|---|---|---|---|
| Uncompressed | 99.2% | 431K | 1× |
| Network Pruning (Han et al., 2015) | 99.2% | 36K | 12.5× |
| ADMM Pruning (Zhang et al., 2018b) | 99.2% | 6.05K | 71.2× |
| Optimal Brain Surgeon (Dong et al., 2017) | 98.3% | 3.88K | 111× |
| **Progressive Weight Pruning** | 99.0% | 2.58K | 167× |

Table 6: Comparisons of weight pruning with quantization results on LeNet-5 for MNIST data set.

| Method | Acc. Loss | No. Para. | Conv No. bits | FC No. bits | Total data size /Compress rate | Total size w. index /Compress rate |
|---|---|---|---|---|---|---|
| Uncompressed | 0.0% | 430.5K | 32 | 32 | 1.7MB | 1.7MB |
| Iterative pruning (Han et al., 2016) | 0.1% | 35.8K | 8 | 5 | 24.2KB / 70.2× | 52.1KB / 33× |
| Learning to share (Ullrich et al., 2017) | 0.2% | – | – | – | – | 10.4KB / 162× |
| **Our Method** | 0.2% | 2.57K | 3 | 2 (3 for output layer) | 0.89KB / **1,910×** | 2.73KB / 623× |

For LeNet-5 model compression, we achieve 167× weight reduction with almost no accuracy loss, which is much higher than prior work under the same accuracy. The prior work Optimal Brain Surgeon (Dong et al., 2017) also achieves a high pruning rate of 111×, but suffers from accuracy drop of around 1% (already non-negligible for MNIST data set).

For other types of DNN models, we have tested the proposed method on the facial recognition application on two representative DNN models (Krafka et al., 2016; Ho, 2016). We demonstrate over 10× weight pruning rate with 0.2% and 0.4% accuracy loss, respectively, compared with the original DNN models.

In summary, the experimental results demonstrate that our framework applies to a broad set of representative DNN models and consistently outperforms the prior work. It also applies to the DNN models that consist of mainly convolutional layers, which are different with weight pruning using prior methods. These promising results will significantly contribute to the energy-efficient implementation of DNNs in mobile and embedded systems, and on various hardware platforms.

Finally, some recent work have focused on the simultaneous weight pruning and weight quantization, as both will contribute to the model storage compression of DNNs. Weight pruning and quantization can be unified under the ADMM framework, and we demonstrate the comparison results in Table 6 using the LeNet-5 model as illustrative example. As can be observed in the table, we can simultaneously achieve 167× weight reduction and use 2-bit for fully-connected layer weight quantization and 3-bit for convolutional layer weight quantization. The overall accuracy is 99.0%. When we focus on the weight data storage, the compression rate is unprecedented 1,910× compared with the original DNN model with floating point representation. When indices (required in weight pruning) are accounted for, the overall compression rate is 623×, which is still much higher than the prior work. It is interesting to observe that the amount of storage for indices is even higher than that for actual weight data.

## 4 RELATED WORK ON DNN WEIGHT PRUNING/MODEL COMPRESSION

The pioneering work by Han et al. (2015) shows that DNN weights could be effectively pruned while maintaining the same accuracy after iterative retraining, which gives 9× pruning in AlexNet and 13× pruning in VGG-16. However, higher compression rates could hardly be obtained as the method remains highly heuristic and time-consuming. Extensions of this initial work apply algorithm-level

improvements. For example, Guo et al. (2016) adopts a method that performs both pruning and growing of DNN weights, achieving $17.7\times$ pruning rate in AlexNet. Dai et al. (2017) applies the evolutionary algorithm that prunes and grows weights in a random manner, achieving $15.7\times$ pruning rate in AlexNet. The Optimal Brain Surgeon technique has been proposed Dong et al. (2017), achieving minor improvement in AlexNet/VGGNet but a good pruning ratio of $111\times$ with less than 1% accuracy degradation in MNIST. The $L_1$ regularization method (Wen et al., 2016) achieves $6\times$ weight pruning in the convolutional layers of CaffeNet. Mao et al. (2017) uses different versions of DNN weight pruning methods, from the fine-grained pruning to channel-wise regular pruning methods. Recently, the direct ADMM weight pruning algorithm has been developed (Zhang et al., 2018b), which is a systematic weight pruning framework and achieves state-of-the-art performance in multiple DNN models.

The above weight pruning methods result in irregularity in weight storage, in that indices are need to locate the next weight in sparse matrix representations. To mitigate the associated overheads, many recent work have proposed to incorporate regularity and structure in the weight pruning framework. Representative work include the channel pruning methods (He et al., 2017; Mao et al., 2017), and row/column weight pruning method (Wen et al., 2016). The latter has been extended in a systematic way in Zhang et al. (2018c). These work can partially mitigate the overheads in GPU, embedded systems, and hardware implementations and result in higher acceleration in these platforms, but typically cannot result in higher pruning ratio than unrestricted pruning. We will investigate the application of progressive weight pruning to the regular/structured pruning as future work.

## 5 CONCLUSION

This work proposes a progressive weight pruning approach based on ADMM, a powerful technique to deal with non-convex optimization problems with potentially combinatorial constraints. Motivated by dynamic programming, the proposed method reaches extremely high pruning rates by using partial prunings, with moderate pruning rates in each partial pruning step. Therefore, it resolves the accuracy degradation and long convergence time problems when pursuing extremely high pruning ratios. It achieves up to $34\times$ pruning rate for the ImageNet data set and $167\times$ pruning rate for the MNIST data set, significantly higher than those reached by work in the existing literature. Under the same number of epochs, the proposed method also achieves better convergence and higher compression rates.

### ACKNOWLEDGMENTS

Financial support from the National Science Foundation under awards CAREER CMMI-1750531 and ECCS-1609916 is gratefully acknowledged.

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
