# OpenReview forum: "Progressive Weight Pruning Of Deep Neural Networks Using ADMM"
_ICLR.cc/2019/Conference_

### Official Review · AnonReviewer3 · 2018-10-30
**Good compression rate of weights empirically, but lack of idea novelty**

**Rating:** 4
**Confidence:** 4

**Review:**

This paper proposed a progressive weight pruning approach to compress the learned weights in DNN. My major concerns about the paper are as follows:

1. Novelty: The proposed approach heavily relies on the one in (Zhang et. al. 2018b) as shown in Sec. 2.2 for 1 page, making the paper as being an incremental work, like finding better initialization for (Zhang et. al. 2018b).

2. Faster convergence: First of all, from Fig. 3 I do not believe that both methods converged, as both performances vary a lot with significant gaps. In terms of being faster, I do not think that it makes sense by comparing numbers of epochs in training with only one approach. There is no theoretical or empirical evidence (e.g. running time) to support this claim.

3. I do not understand how the proposed approach is motivated by DP. To me it is more like a greedy search algorithm, while DP has the ability to locate global maximum. Does the proposed approach guarantee to find the maximum accuracy? Also, in Fig. 2 why was the best partial model replaced with the new one, rather than the worse one? There is no explanation to this at all. Besides, I do think this approach is very heuristic, same as some other approaches in the related work.

4. Experiments: Since the performance varies a lot as shown in Fig. 3, how are the numbers calculated? Average? Best one? With/without cross-validation to tune parameters? How much gain in terms of running time in testing can you get with more compact models in practice? A training/testing behavior analysis is highly appreciated.

---

> ### Author Response · Authors · 2018-11-27
> **Thanks for your comment. We revise our manuscript and we like to share our updates with you**
>
> (Novelty) We study multiple existing pruning methods and realize that performance of pruning methods degrade severely when targeting very high pruning rates. Our main contribution is to identify the degradation behavior and provide a progressive way of pruning to mitigate this issue. To better support our claim, we add Figure 3 in our revised manuscript to demonstrate that performance degradation in high pruning rates are common for multiple pruning methods. Our experiments on large networks such as VGG, ResNet-50 for dataset such as ImageNet, ResNet-56 for dataset like Cifar-10 confirm that degradation will happen and progressive way of pruning can well mitigate it.
>
> As solving DNN is a non-convex problem, we think that progressively solving it is a good approach. Our recent search shows that. Independent from our work, a paper from ICLR 2018 also discovered this approach  “Progressive Growing of GANs for Improved Quality, Stability, and Variation”
>
>
> (why it’s DP?)
> We want to clarify that our algorithm is not dynamic programming. As we stated in the paper, we got inspired by dynamic programming. Our experiments suggest that extremely high pruning rates cannot be achieved by one step without suffering significant accuracy loss. The key question is how to find the best intermediate steps.  Like similar observation from other independent work, we propose a progressive way of solving a difficult non-convex problem and our results prove that it’s very effective. The underlying reason why progressive way of solving problem is effective needs more attention and we don’t want to claim too early whether this progressive weight pruning is dynamic programming or greedy algorithm. However, keeping multiple good intermediate results is a stable, safe way to achieve high pruning rates in our case.
>
> (Clarification of Figure 3)
> In our revised manuscript, we replace the original Figure 3 with a new figure. We believe this new Figure 3 better demonstrates the key insight and contribution of our paper: Limitation of existing pruning methods in high pruning rates and progressive way of pruning can well mitigate the issue. We compare our methods with multiple existing pruning methods to support our claim.
>
> (Tuning hyper-parameter)
> Our hyper-parameter is decided in design time and we don’t use validation set for tuning.
>
> (Running time in more compact model)
> Although the focus of this paper is not the speed up of the running time, we do like to share our results on running time speed up.
> Unlike existing pruning methods, our methods can compress convolutional layers without large performance degradation. Therefore, with minor accuracy loss, we can prune convolutional layers by 13.2X and obtain 10.2X running time speed up in Intel i7 CPU,  20X running time speed up in Rasperry Pi.

---

### Official Review · AnonReviewer2 · 2018-10-31
**Concerns about proposed method and experiments**

**Rating:** 5
**Confidence:** 4

**Review:**

This paper focus on weight pruning for neural network compression. The proposed method is based on ADMM optimization method for neural network loss with constraint on the l_0 norm of weights, proposed in Zhang et al. 2018b. Two improvements, masked retraining and progressive pruning, are introduced. Masked retraining set the weights to zero at early stages and stop updating those weights. Progressing pruning keeps a buffer of partial pruning results and select the best performed model for further pruning. The proposed method achieves 30x compression rate for AlexNet and VGG for ImageNet.

I have the following concerns about the proposed method.
- It is unclear to me what is the benefit of ADMM for solving the sparse regularized NN optimization problem. Why is it better than projected gradient descent or proximal gradient method used in previous network pruning? I understand the proposed method is based on Zhang et al. 2018b, but a strong argument will support the draft.
- I fail to understand the claim ``at convergence, the pruned DNN model will not be exactly sparse’’ in section 2.3. Z will always be sparse after the projection step in (5). At convergence, the linear constraint should be satisfied, which makes W = Z to be sparse.
- Please describe the the proposed method in detail. The current description is very vague and I do not think it can be reimplemented based on the current draft. In each outer loop of ADMM, (4)(5) and dual update is applied (I consider solving (4) is the inner loop). How is the mask generated and fit into these equations? For progressive pruning, it looks to me there is an outer loop outside the outer loop of ADMM. Please provide details on how many iterations, and how the compression rate is decided for each iteration.
- The hyper-parameters of the proposed method is unclear. It is a bit strange the optimization parameter \rho could control the pruning rate (section 3.1). As described before, I guess the proposed method has three loops. How is the iterations counted, like for Figure 3. Please clarify the experiments are fair comparison, the better results are not because of more weight updates from the three loops.
- It is unclear what is the benefit of masked retraining. It looks to me this kind of greedy approach will harm the performance (I have to guess if a weight is masked to be zero, it will never be updated or recovered). What happens if there are a lot of weights (Z in (5)) are zero at the early stage?
- The progressive pruning looks heuristic and I am not convinced the buffer is necessary. There is always a progressive pruning trace that can directly lead to the results without selecting from candidates. For example, in Figure 2, we can just train model from 15x to 24x to 30x.
- The following works are related.
Li et al. Pruning Filters for Efficient ConvNets. ICLR 2017
Alvarez et al. Learning the number of neurons in deep networks. NIPS 2016

=============== after rebuttal ===================
I appreciate the authors' feedback and slightly raise the score.

Though the compression results look good, I still have some concerns about the method. The motivation of the proposed method is not strong. The proposed mask is greedy and sounds ad-hoc. The proposed progressive pruning looks expensive.

The proposed method looks time consuming. For the experiments, I would love to see the training time comparing with baselines in table 1, not only the ADMM method in table 2. A fair comparison could be wall-clock time, or number of gradient updates for neural networks.

---

> ### Author Response · Authors · 2018-11-27
> **Thanks for your comment. We revise our manuscript and we like to share our updates with you**
>
> (1) Benefit of ADMM and why it’s better than PGD:
> The reason ADMM performs better than methods such as PGD is that using ADMM, we solve a dynamic DNN regularization problem (in Eqn. (4)) in each ADMM iteration, in which the regularization target is analytically adjusted. This aspect is lacking in methods such as PGD.
>
> (2) Claim at convergence(why W is not exactly sparse after ADMM?)
> At Eqn. (5), the optimization variable is Z and thus W will not be exactly sparse after solving this subproblem. W will be sparse after mapping to Z.
>
> (3) (Please describe the proposed method in detail..)
> We added pseudo code for our algorithm in our revised manuscript. We release our code both in Tensorflow and Caffe. It is actually easy to reproduce and we have received feedbacks from multiple institutions that they have successfully tested the models and code.
>
> For generation of mask, we define the mask in the beginning of the process. By projecting W to set S, we obtain a mask in which values are 0 for zero weights and 1 for non-zero weights.
> When applying mask during training, we multiply the gradients of weights with corresponding mask values to derive the actual gradients for weight updating.
>
> As shown in the Algorithm 1 of our revised manuscript, (4) (5) (6) are all in the inner loop. The “outer loop” is the number of intermediate (partial) pruning results to achieve our goal. In practice, it suffices to take two steps to achieve best result (e.g., 1X->15X->30X). The higher performance is not resulted from unfairness. We compare results in Table 2 by using the same overall training time for both methods.
>
> (4) (Why hyper-parameter rho could control the pruning rate?)
> We want to clarify that rho does not control the pruning ratio. Empirically, making rho slightly higher when targeting high pruning rate gives slightly better results. We provide that hint for people who want to reproduce our results.
>
> (5) (Doesn’t mask retrain harm the performance?)
> After the ADMM step, neural networks are well prepared for weights (in our extremely high pruning rate cases, a lot of weights) to be very close to 0. However, since those weights are not exactly sparse, the mapping from weights close to zero to exactly zero will make the networks lose some accuracy. The masked retrain encourages the NN to update only nonzero weights so that it can recover from the mapping step. Therefore, it will not harm the performance (accuracy).
>
> (6)(Why we like to keep the buffers?)
>
> We want to clarify that those partial pruned results are themselves state-of-art pruned models in their own pruning rates, saved to the disk, instead of buffers in memory. The meaning of having intermediate results generated is two-folds. In real-world scenario, those models can be immediately used and deployed in product line. Also, even though in our experiments we can quickly find a clear trace, for example, our best result such as 30X compression for AlexNet is obtained by 0->15X->30X.
> The key contribution of this work is that we discover existing pruning methods have performance degradation when pursuing high pruning rates and progressive way of pruning can mitigate this degradation. Independent from our work, a ICLR 2018 paper  “Progressive Growing of GANs for Improved Quality, Stability, and Variation” has the similar discovery.
> The underlying reason why progressive way of solving problem can mitigate the performance degradation needs more attention and we want to be cautious in our work. Those candidates for intermediate results all have potential to achieve very high pruning rates. That’s why we want to keep a few of them in each round of progressive pruning.
>
> (7)
> Thanks for providing the related work to compare. We compare our method with Li et al. and show our method largely outperforms it.
> ResNet-56 for Cifar-10
> Method			Pruning rate		Accuracy
> Li et al				1.16X			93.06%
> Our method			2.00X			93.19%

---

### Official Review · AnonReviewer1 · 2018-11-02
**The paper proposes a progressive pruning technique which imposes structural sparsity constraint on the weight parameter. Since solving the minimization with sparsity constraint is hard in general, the paper rewrites the optimization as an ADMM framework. While ADMM method suffers from slow convergence, a progressive weight pruning approach is proposed, which falls into curriculum learning.**

**Rating:** 5
**Confidence:** 3

**Review:**

The authors argue that ADMM-based approach achieves higher accuracy than projected gradient descent. However, experimental evidence is lacking. The authors should compare to a trivial variant of Adam that a projection step is followed by the gradient update.

Experimental results are weak. It seems that the proposed method only works on small networks such as AlexNet and LeNet. On larger networks such as VGG-16 and ResNet, the proposed method achieves higher compression rates at the expense of lower accuracies compared to the related works. Thus, the authors should compare with other methods with the same compression rates.

As ADMM is sensitive to the penalty parameter, the authors should also conduct more experiments to show robustness of the choice of the penalty parameter across different experiments.

---

> ### Author Response · Authors · 2018-11-27
> **Thanks for your comment. We revise our manuscript and we like to share our updates with you.**
>
> (whether we get better pruning rates in exchange of accuracy drop?)
>
> In our revised manual, Table 4 compares our results on ResNet-50 with other methods.
> Table A (result on ResNet-50 for ImageNet)
> Method				                              Top5-Accuracy	       Pruning rates
> Uncompressed 				                           92.4%			1X
> Fine-grained Pruning (Mao et al., 2017)	           92.3%			2.6X
> Our method                                                             92.3%                  6X
> Our method					                           92.1%			9.16X
> We also compare our method with Li et al. Pruning Filters for Efficient ConvNets. ICLR 2017. Here we compare our results on ResNet-56 with Li’s.
> Table B (result on ResNet-56 for Cifar-10)
> Method				Pruning rate		Accuracy
> Li et al					1.16X			93.06%
> Our method				2.00X			93.19%
> Table C (result on VGG-16 for ImageNet)
> Method				 Pruning rate		Top-5 Accuracy
> Optimal Brain Surgeon      13.3X			89.0%
> Our method			       13.3X			89.2%
>
> Also Table 2 in the revised manuscript shows that notable accuracy increase can be observed with a higher pruning rate than prior work.
>
> Our new results show that we can convincingly do better in networks such as ResNet-50, ResNet-56 and VGG-16. Moreover, the main contribution of this paper is that we find that many pruning methods incur accuracy degradation in high pruning rates. In the first version of our paper, we mostly use Table 2 to demonstrate how performance of Zhang et. al. 2018b degrades rapidly when compression rate gets very high and how progressive way of pruning mitigates the degradation. In our revised manuscript, we add Figure 3 to support our claim: the degradation is common to multiple pruning methods and the proposed method mitigates the degradation.
>
> (Hyperparameter rho) We use very close rho (1.5e-3) for all networks we tested such as LeNet , AlexNet, VGG-16, ResNet-18/ResNet-50/ResNet-56. The pruning results are not sensitive to the choice of rho. When the choice of rho ranges from 1e-2 to 1e-4, the results do not significantly change. The result will change significantly if rho is decreased and increased by larger orders of magnitude.

---

### Meta-Review · Area_Chair1 · 2018-12-17
**incremental work**

**Confidence:** 5
**Recommendation:** Reject

**Metareview:**

The paper proposes a progressive pruning technique that achieves high pruning ratio. Reviewers have a consensus on rejection. Reviewer 1 pointed out that the experimental results are weak. Reviewer 2 is also concerned about the proposed method and experiments. Reviewer 3 is is concerned that this paper is incremental work. Overall, this paper does not meet the standard of ICLR. Recommend for rejection.